# Comparative Analysis of Porcine Adipose- and Wharton’s Jelly-Derived Mesenchymal Stem Cells

**DOI:** 10.3390/ani13182947

**Published:** 2023-09-17

**Authors:** Ga Yeon Kim, Gyu Tae Choi, Jinryong Park, Jeongeun Lee, Jeong Tae Do

**Affiliations:** 1Department of Stem Cell and Regenerative Biotechnology, KU Institute of Technology, Konkuk University, Seoul 05029, Republic of Korea; poibe0605@naver.com (G.Y.K.); ch08david@naver.com (G.T.C.); wlsfyd1321@naver.com (J.P.); 23D Tissue Culture Research Center, Konkuk University, Seoul 05029, Republic of Korea; 3Department of Agricultural Convergency Technology, Jeonbuk National University, Jeonju 54896, Republic of Korea; dlwjddms0625@naver.com

**Keywords:** porcine, mesenchymal stem cell, proliferation, differentiation, oxygen consumption rate

## Abstract

**Simple Summary:**

Mesenchymal stem cells (MSCs) are important stem cells that have potential for use in cultured meat as well as for clinical applications. Among various animal species, porcine MSCs have received comparatively less attention. In this study, we aimed to compare two types of porcine MSCs by comparing proliferation rate, differentiation potential and mitochondrial metabolism. Adipose-derived stem cells showed better adipogenic and chondrogenic differentiation potential, higher proliferative capacity, and higher mitochondrial oxygen consumption than Wharton’s jelly-derived mesenchymal stem cells. This comparative analysis will be useful for understanding porcine MSCs, which are rarely studied in pigs.

**Abstract:**

Mesenchymal stem cells (MSCs) are promising candidates for tissue regeneration, cell therapy, and cultured meat research owing to their ability to differentiate into various lineages including adipocytes, chondrocytes, and osteocytes. As MSCs display different characteristics depending on the tissue of origin, the appropriate cells need to be selected according to the purpose of the research. However, little is known of the unique properties of MSCs in pigs. In this study, we compared two types of porcine mesenchymal stem cells (MSCs) isolated from the dorsal subcutaneous adipose tissue (adipose-derived stem cells (ADSCs)) and Wharton’s jelly of the umbilical cord (Wharton’s jelly-derived mesenchymal stem cells (WJ-MSCs)) of 1-day-old piglets. The ADSCs displayed a higher proliferation rate and more efficient differentiation potential into adipogenic and chondrogenic lineages than that of WJ-MSCs; conversely, WJ-MSCs showed superior differentiation capacity towards osteogenic lineages. In early passages, ADSCs displayed higher proliferation rates and mitochondrial energy metabolism (measured based on the oxygen consumption rate) compared with that of WJ-MSCs, although these distinctions diminished in late passages. This study broadens our understanding of porcine MSCs and provides insights into their potential applications in animal clinics and cultured meat science.

## 1. Introduction

Mesenchymal stem cells (MSCs) are multipotent and differentiate into various mesodermal lineages such as adipose, bone, cartilage, and connective tissues. Therefore, MSCs reside in a variety of mesodermal tissues and can be derived from the adipose tissue, umbilical cord, bone marrow, and other mesodermal tissues [1,2,3]. MSCs exhibit a consistent spindle-shaped morphology and share common surface marker profiles, regardless of their origin, as defined by the International Society for Cell Therapy (ISCT) [4]. The prevalent markers of MSCs include CD73, CD90, and CD105, and they are typically negative for hematopoietic markers such as CD14, CD34, and CD45 [4,5,6]. However, MSCs differ in their proliferative capacity, differentiation potential, molecular signature, and signaling pathways depending on their specific tissue of origin and individual characteristics [7,8,9]. A previous study suggested that adipose-derived stem cells (ADSCs) show superior adipocyte differentiation potential compared with that of bone marrow-derived MSCs (BMSCs) [1,10]. Moreover, BMSCs demonstrate a tenfold increase in osteogenic gene expression compared with that of Wharton’s jelly-derived mesenchymal stem cells (WJ-MSCs) [1,10]. In addition, WJ-MSCs tend to differentiate into chondrocytes and diminish their potential for adipogenic differentiation [7]. ADSCs of the same origin exhibit variations in differentiation efficiency depending on the specific transplantation site in vivo. For example, when ADSCs are transplanted near the heart, they differentiate favorably into cardiovascular tissues [11,12]. On the contrary, upon transplantation into joints, both WJ-MSCs and ADSCs showed favorable differentiation into cartilage [9], suggesting that MSC differentiation is influenced by both the cell source and surrounding environment. MSCs exert characteristics such as immunosuppression, easy extraction, easy expansion, and plasticity (multilineage differentiation ability). Therefore, MSCs have garnered considerable interest in cell therapy and cultured meat research.

Recently, MSCs have been the subject of several active studies; however, few studies have focused on a comparison between MSCs extracted from pigs. The purpose of this study was to compare the differences in the proliferative capacity, direction of differentiation, and metabolic capacity of porcine MSCs derived from different tissues. We selected two different types of MSCs, ADSCs, and WJ-MSCs, and investigated their characteristics and correlation between differentiation, proliferation, and metabolic ability to contribute to a deeper understanding of porcine MSC characteristics.

## 2. Materials and Methods

### 2.1. Isolation and Culture of Porcine Wharton’s Jelly MSCs (WJ-MSCs) and Adipose Derived-Stem Cells (ADSCs)

ADSCs were extracted from the back fat of 1-day-old piglets (n = 3), and WJ-MSCs were extracted from umbilical cords (n = 20). Wharton’s jelly and adipose tissue were washed two to three times using phosphate-buffered saline (PBS; Welgene, #LB004-02) containing 10% antibiotic–antimycotic solution (A/A; Gibco, #15240062). The tissues were minced as much as possible and digested with 0.2% collagenase type II (Worthington, #LS004176) in DMEM/F12 (Gibco, #11320-033) with shaking at 100 rpm and 37 °C for 60 min. The digestion medium was neutralized using MEM alpha (gibco, #12571071) supplemented with 10% fetal bovine serum (FBS, cytiva, #SH30071.03) and 1% A/A. The neutralized medium was filtered using a 100-µm cell strainer (Falcon, #352360). The mixture was centrifuged at 250× *g* (1100 rpm) for 5 min at room temperature, and the cell pellet was resuspended in MEM alpha supplemented with 10% FBS, 1% penicillin–streptomycin–glutamine (PSG, Gibco, #10378-016), and basic fibroblast growth factor (bFGF, Gibco, #13256-029, 10 ng/ml). After isolation, the ADSCs and WJ-MSCs were cultured in an incubator at 37 °C in a humidified 5% CO_2_ atmosphere until they reached 90% confluence (Figure 1A). Subsequent passaging was carried out using 0.25% trypsin-EDTA solution (TE, Gibco, #25200072), followed by seeding the cells in a 100 mm culture dish at a density of 1 × 10^6^ cells/dish. Cells were cultured until passage 20. Frozen stocks of 1 million cells/cryotube were preserved in liquid nitrogen. The cells at passage 3 were termed the “early passage” cells, and cells at passage 20 were termed as “late passage” cells.

### 2.2. Flow Cytometry

ADSCs and WJ-MSCs were detached using 0.25% trypsin EDTA (Gibco), and the collected cells were washed with PBS containing 1% BSA. The cells were divided into 1 × 10^6^ cells and incubated for 1 h at 4 °C with primary antibodies against CD34 (Bioss, Beijing, China), CD45 (Proteintech, Rosemont, IL, USA, clone number. 4E9B2), CD73 (R&D Systems, Minneapolis, MN, USA), CD90 (R&D Systems, Minneapolis, MN, USA), and CD105 (Novus Biologicals, Centennial, CO, USA, clone number. MEM-229). The cells were then washed with PBS containing 1% BSA and incubated for 1 h at 4 °C with secondary antibodies against PE-conjugated anti-rabbit (R&D Systems, MN, USA), PE-conjugated anti-sheep (R&D Systems, MN, USA), and APC-conjugated anti-mouse (R&D Systems, MN, USA) antibodies. Appropriate secondary antibodies were used according to the primary antibodies. Unlabeled cells and those labelled only with the secondary antibody were used as negative controls. Subsequently, the cells were washed with PBS containing 1% BSA. The fluorescence of the stained samples was analyzed using a FACSCalibur flow cytometer (BD Biosciences) and BD CellQuest Pro software v9.

### 2.3. Cell Proliferation Analysis

ADSCs and WJ-MSCs were seeded into individual 6-well plates at a density of 1 × 10^5^ cells/well. The seeding process was repeated thrice consecutively for each cell type (n = 3). The culture media were refreshed every 24 h, and the cells were cultured for 72 h. After 72 h of culture, the cells in each well were detached using 0.25% trypsin-EDTA (Gibco, #25200072). The cell counts were determined using an inverted microscope equipped with a hemocytometer.

### 2.4. Cell Counting Assay Using Kit-8 (CCK-8)

ADSCs and WJ-MSCs were seeded into 96-well plates at a density of 1 × 10^3^ cells/well (n = 10). The cells were then treated with Cell Counting Kit-8 (CCK-8, Dojindo, #CK04-11) solution following the manufacturer’s instructions and incubated at 37 °C for 4 h. The OD of each well was measured using a microplate reader at a wavelength of 450 nm.

### 2.5. Immunocytochemistry

For immunochemistry, the cells were cultured until 70–80% confluency was achieved. The cells were fixed in 4% paraformaldehyde at 4 °C for 30 min. Following fixation, the cells were washed with PBS and treated with 0.3% Triton X-100 in PBS for 10 min. Subsequently, a blocking step was performed in PBS containing 3% bovine serum albumin (Bovogen, BSAS0.1) at 1 h at 25 °C. The blocked cells were then treated overnight with the Ki67 (1:200; GeneTex, GTX16667) primary antibodies at 4 °C. After 16 h, the primary antibodies were eliminated by washing with PBS for 10 min, and the cells were labelled with fluorescent secondary antibodies (1:500, Abcam, Alexa Fluor 488). Finally, the cells were washed and treated with DAPI in 0.3% Triton X-100 in PBS for 4 min at 25 °C and washed (n = 4).

### 2.6. Adipogenic, Chondrogenic, and Osteogenic Differentiation and Staining

ADSCs and WJ-MSCs were seeded on 60 mm tissue culture dishes at a density of 3 × 10^5^ cells for adipogenic and chondrogenic differentiation. The adipogenic differentiation media consisted of DMEM low glucose containing 10% FBS, 1% PSG, 1 µM dexamethasone, 500 µM IBMX, 10 µg/ml human insulin, and 100 µM indomethacin. The chondrogenic differentiation medium comprised DMEM low glucose containing 10% FBS, 1% PSG, dexamethasone 100 nM, 50 µg/mL L-Ascorbic acid 2-phosphate, and 10 ng/ml TGF-β1. Osteogenic differentiation was induced using the StemMACS OsteoDiff medium. Cell density determination and seeding were performed according to the manufacturer’s instructions. The differentiation medium was replaced after 72 h. Adipogenic differentiation was assessed at 21 days using Oil Red O staining solutions. Chondrogenic differentiation was executed for 21 days, while osteogenic differentiation was performed for 14 days. Alcian blue and Alizarin red were used to evaluate chondrogenic, and osteogenic differentiation, respectively. All staining procedures were performed according to the manufacturer’s instructions.

### 2.7. RNA Isolation and RT-qPCR

RNA extraction was performed using the TRIzol reagent (Invitrogen, 15596026, Carlsbad, CA, USA) according to the appropriate protocol. Then, cDNA was synthesized from 1 µg total RNA using SuperScriptTM III Reverse Transcriptase (Invitrogen, Waltham, MA, USA, 18080-044), 10 mM dNTP Mix (Invitrogen, 18427-013), and Oligo (dT) 12–18 Primer (Invitrogen, 18418-012). Real-time Quantitative polymerase chain reaction (RT-qPCR) was performed using TOPrealTM qPCR 2X PreMIX (Enzynomics, Daejeon, Republic of Korea, RT500M). The results were analyzed using a Roche LightCycler 5480 (Roche). The thermal cycles comprised 50 cycles at 95 °C for 10 s, 60 °C for 15 s, and 72 °C for 20 s. The primers used for RT-qPCR are listed in Table 1.

### 2.8. Oxygen Consumption Rate Analysis

The oxygen consumption rate (OCR) was measured by analyzing the cells using an XFp analyzer (Seahorse Bioscience, Chicopee, MA, USA). Overall, 3.5 × 10^4^ ADSC and 2.0 × 10^4^ WJ-MSC cells were cultured for 24 h after being attached to an XF cell culture miniplate pre-coated with diluted Matrigel (Corning, NY, USA, 356230). Before analysis, the medium was changed to XF Assay Medium supplemented with sodium pyruvate (Agilent, 103578-100), d-glucose (Agilent, Santa Clara, CA, USA, 103577-100), and l-glutamine (Agilent, 103579-100). To measure mitochondrial respiration, the OCR was assessed using oligomycin (1.5 µM), FCCP (0.8 µM), and rotenone/antimycin A (0.5 µM) (Agilent). The assay was performed according to the manufacturer’s instructions.

### 2.9. Statistical Analysis

All experiments were performed using the SAS software v9.4 (SAS Institute Inc., Cary, NS, USA). The significance of the differences was determined using t-test or analysis of variance (ANOVA) with Duncan’s Multiple Range Test for post hoc multiple comparisons. The data are presented as mean ± standard deviation (SD).

## 3. Results

### 3.1. Characterization of Porcine ADSCs and WJ-MSCs

At early passages (passage 3), both ADSCs and WJ-MSCs showed a fibroblast-like spindle shape (Figure 1B). During the late passage (passage 20), they became elongated and flattened while retaining a morphology similar to that observed in the early passage (Figure 1B). We performed flow cytometry (FACS) to characterize the MSCs and analyze MSC surface markers expression. Both ADSCs and WJ-MSCs highly expressed the positive markers (CD73, CD90, and CD105) but did not express the negative markers (CD34 and CD45) (Figure 1C). Notably, CD73 and CD105 expression in WJ-MSCs was lower than that in ADSCs (Figure 1C). This was corroborated by the results of the RT-qPCR analysis of MSC markers, where negative markers (CD34 and CD45) were not expressed, whereas positive markers (CD44, CD73, CD90, and CD105) were expressed relatively highly compared to negative markers (Figure 1D). CD marker mRNA expression in late passages was consistent with early passages, with positive markers relatively highly compared to negative markers expressed and negative markers were not expressed (Figure 1E).

### 3.2. Comparison of Proliferation Rate between ADSCs and WJ-MSCs

ADSCs consistently exhibited a higher proliferation rate than that of WJ-MSCs across all passages, and both groups showed a gradual decline in proliferation until passage 20 (Figure 2A). This decreasing pattern was further supported by CCK-8 analysis (Figure 2B). In the early stages, there was a significant difference in the absorbance (the evaluation criterion for proliferative capacity) between ADSCs and WJ-MSCs. However, at passage 15, the absorbance of ADSCs was lower than that of WJ-MSCs, but it increased again at later passages. Ki67 immunocytochemistry analysis also confirmed the differences in proliferation rates between ADSCs and WJ-MSCs during early passages. Nevertheless, no significant differences were observed as the passages progressed (Figure 2C,D).

### 3.3. Characterization of Differentiation Potential of ADSCs and WJ-MSCs

Given that cell origin affects the MSC differentiation potential [7,8,9], we compared the differentiation potentials between ADSCs and WJ-MSCs. As the differentiation potential may differ between early and late passages, cells from passages 3 and 20 were grouped and subjected to differentiation. ADSCs and WJ-MSCs from both early and late passages were differentiated to produce three distinct cell types: adipocytes, chondroblasts, and osteoblasts. Lineage differentiation was validated using Oli Red O, Alcian blue, and Alizarin red S staining to indicate adipocytes, chondroblasts, and osteoblasts, respectively (Figure 3A). In both the early and late passages, ADSCs predominantly differentiated into adipocytes, whereas WJ-MSCs rarely differentiated into adipocytes. Chondrogenic and osteogenic differentiation was observed in both ADSCs and WJ-MSCs. To gain a better understanding of the differences between the various ADSCs and WJ-MSCs lineages, RT-qPCR analysis was conducted to assess the expression levels of lineage-specific markers after the induction of differentiation (Figure 3B–G). Adipogenic markers, PPARγ (Peroxisome Proliferator-Activated Receptor γ), C/EBPα (CCAAT/Enhancer-Binding Protein α), FAS (Fatty Acid Synthase), FABP3 (Fatty Acid Binding Protein 3), FABP4 (Fatty Acid Binding Protein 4), and GLUT4 (Glucose Transporter Type 4) were substantially upregulated in ADSCs following induced adipogenic differentiation (Figure 3B). In contrast, no significant differences were observed in the expression of adipogenic markers between WJ-MSCs before and after differentiation (Figure 3B). These patterns persisted in the late-passage samples (both in ADSCs and WJ-MSCs) (Figure 3B,C). 

We compared the chondrogenic differentiation using two markers: Collagen type I alpha 1 chain (COL1A1) and Collagen type II alpha 1 chain (COL2A1) (Figure 3D,E). RT-qPCR analysis revealed that ADSCs showed higher potential for chondrogenic differentiation than that of WJ-MSCs (Figure 3D), and this distinction persisted in early and late passages (Figure 3D,E). 

Finally, the osteogenic differentiation potential was assessed using the osteogenic markers, Runt-related transcription factor 2 (RUNX2) and Distal-Less homeobox 5 (DLX5) (Figure 3F,G). RT-qPCR analysis revealed that no significant difference was observed in the early passages between ADSCs and WJ-MSCs in RUNX2 expression prior to the induction of differentiation. However, RUNX2 expression was more than 20-fold upregulated in differentiated WJ-MSCs (Figure 3F). In the late passages, a significantly higher expression of RUNX2 and DLX5 was observed in differentiated WJ-MSCs (Figure 3G). Collectively, these results indicate that regardless of passage status, ADSCs predominantly differentiated into adipogenic and chondrogenic lineages, whereas WJ-MSCs displayed a high differentiation potential into the osteogenic lineage.

### 3.4. Comparative Analysis of Oxidative Phosphorylation in ADSCs and WJ-MSCs

The examination of mitochondrial metabolic activity is crucial for understanding the mechanisms by which cells produce and utilize energy. To better understand the differences between the two cell types, we compared their mitochondrial energy metabolism, which is crucial for stem cells and differentiation [13]. Exploring the discrepancies in the OCR, which is a metric that reflects oxidative phosphorylation (OXPHOS) activity, in ADSCs and WJ-MSCs in both the early and late passages, we found that ADSCs displayed a three-fold higher basal respiration rate than those of WJ-MSCs at early passages (Figure 4A,B); however, no significant differences were observed in the late passages (Figure 4F,G). Maximal respiration rate was not significantly different between ADSCs and WJ-MSCs during both early and late passages (Figure 4C,H). In terms of ATP-producing coupled respiration, ADSCs showed three-fold higher ATP production than those of WJ-MSCs in the early passage, although this difference was not observed in the late passage (Figure 4D,I). Additionally, WJ-MSCs showed approximately twice the spare respiratory capacity of ADSCs in the early passage; however, this difference was not significant in the late passage (Figure 4E,J). Moreover, the difference in OXPHOS between ADSCs and WJ-MSCs was distinct during the early stages but became less significant as the passages increased. This may be because both cell types adapted to the in vitro environment during long-term culture and acquired similar energy metabolisms.

## 4. Discussion

To confirm the successful establishment of MSCs, we used positive CD markers such as CD44, CD73, CD90, and CD105, which are important surface proteins for regulating differentiation [14,15]. The expression of CD surface markers on MSCs is specific and varies depending on the tissue source [16,17]. Additionally, the mRNA levels of CD markers may vary with the passage of MSCs [15,16]. We found that CD44, CD73, and CD105 were differentially expressed in ADSCs and WJ-MSCs derived from different tissue sources. Notably, the expression pattern of CD44 changed during prolonged culture, indicating that CD marker expression patterns can change during prolonged culture. 

CD44, also referred to as P-glycoprotein 1, is a receptor for hyaluronan or hyaluronic acid and plays a role in the inhibition of osteogenic differentiation [18,19] and the induction of chondrogenic and adipogenic differentiation [20,21,22]. The results of the current study also showed the upregulation of CD44 expression in both ADSCs and WJ-MSCs. Notably, the CD44 expression level during early passages was higher for ADSCs than for WJ-MSCs. However, the opposite pattern was observed during late passages. CD73, also known as ecto-5′-nucleotidase [23], is associated with the differentiation and promotion of cartilage and bone [14,24]. However, despite the high CD73 expression in ADSCs, they rarely differentiated into the osteogenic lineage. CD105 is a co-receptor of the TGF-beta family and is associated with chondrogenesis by regulating the smad2/3 and smad1/5 pathways [25,26]. Decreased expression of CD105 promotes osteogenic differentiation and favors adipogenic differentiation [14,27]. In the current study, elevated CD105 expression was observed in WJ-MSCs, resulting in proficient osteogenic differentiation; however, chondrogenic and adipogenic differentiation was limited. Overall, the expression patterns of these CD markers in undifferentiated MSCs do not necessarily ensure their ability to differentiate into specific lineages [14,28]. 

During in vitro MSC culture, the aging process is accelerated compared with that in the in vivo environment [29]. Here, we observed that the proliferation rates of ADSCs and WJ-MSCs were significantly higher in early passages compared to late passages. However, as the passages progressed, the proliferation rate decreased, and the proliferating capacities of the two cell lines became similar. These findings support those of previous studies on the proliferative capacity of MSCs, which showed functional impairments such as reduced proliferation and telomere shortening due to aging during long-term culture [30,31,32]. Additionally, during long-term culture with aging, the cells show reduced proliferation, differentiation, and metabolic capacity [33]. Our results align with these observations, as we found that cells in later passages exhibited reduced proliferation, differentiation, and metabolic capacities compared with that of their earlier passage counterparts.

We propose that ADSCs show a strong propensity for adipogenic differentiation but display a limited capacity for bone tissue differentiation. Conversely, WJ-MSC showed diminished adipogenic differentiation potential but high bone differentiation potential. Previous studies have suggested that the presence of fibroblast growth factor 2 (FGF2) can reduce the efficiency of adipogenic and chondrogenic differentiation [34,35,36,37]. During MSC differentiation into adipogenic, chondrogenic, and osteogenic lineages, FGF2 is removed from the culture medium, resulting in efficient adipogenic and chondrogenic differentiation of ADSCs. This result indicates that the antagonistic effect of FGF2 was reset upon the initiation of the differentiation processes. Additionally, in comparisons between MSCs from different species, it was observed that WJ-MSCs exhibited a low capacity for adipogenesis [7,10,38]. In particular, human WJ-MSCs were reported to have significantly lower capacity for adipogenic differentiation [39,40]. Our findings in pigs align with this pattern. 

In this study, PPARγ, C/EBP, FAS, FABP3, FABP4, and GLUT4 were used as adipogenic differentiation markers. Most of these markers except for FABP4 were highly expressed in undifferentiated ADSCs compared with that in undifferentiated WJ-MSCs. This may explain why ADSCs are proficient at differentiating into adipogenic lineages. FAS and FABP4, which are involved in lipid signaling, play a role in proliferation and cell survival [41]. Expression of PPARγ in fibroblasts or muscle cells induced adipose differentiation [42,43]. PPARγ suppresses cancer cell growth via the ERK pathway and induces apoptosis [44,45]. Therefore, high PPARγ expression in cells induces adipocyte differentiation. FABP3, which is activated by PPARγ, is related to fatty acid metabolism in the heart and skeletal muscle and has been reported to act as a chaperone that regulates the availability, mobility, and utilization of fatty acids [46,47]. High OXPHOS activity is observed when fatty acids are elevated [48]. In addition, in rapidly proliferating cell, the expression of FAS (Fatty acid synthase)-related genes increases, resulting in fatty acid enrichment [49]. As fatty acids increase, GLUT4 is activated through mTOR and AMPK pathways, resulting in high OXPHOS activity [48,50]. These observations are consistent with our results showing that ADSCs display higher basal and maximal respiration rates than those of WJ-MSCs (Figure 4A–C).

The current study revealed higher expression of PPARγ in undifferentiated ADSCs than in WJ-MSCs. However, PPARγ expression affects proliferation and metabolism by activating other adipogenic markers such as FAS and FABP3 [46,48,49]. These results show that increased PPARγ expression induces the upregulation of downstream adipogenic genes, including FAS and FABP3, resulting in the rapid proliferation of ADSCs during their early passages. 

Mitochondrial metabolism regulates various cellular processes, such as proliferation, oncogene activation, apoptosis, and reactive oxygen species (ROS) generation [33,51]. Mitochondria consume more oxygen during rapid cell proliferation [52]. High proliferation and high levels of oxygen supply mean that high levels of glucose uptake are required [53]. At early passage, MSCs require particularly high levels of glucose utilization [54]. It is known that when MSCs are cultured in vitro, they use glucose at a faster rate than in vivo conditions [55]. Unlike other cells, MSCs display a high dependence on glucose through glycolysis under a basal state to maintain normal metabolic functions [55]. Our findings further elucidate this phenomenon; the superior proliferation rate of ADSCs compared with that of WJ-MSCs in early passages can be attributed to the elevated basal respiration rate and consequent higher oxygen consumption exhibited by ADSCs.

A rather interesting interpretation of the seahorse data has been raised in the MSC [56]. Enhanced glucose uptake leads to expression of STAT1-dependent indoleamine 2,3 dioxygenase (IDO), an anti-inflammatory factor [54,56]. In our results, the cause of high glucose in ADSCs is explained by the preceding results. Additionally, the high glucose utilization of MSCs provides insight into immunomodulatory due to enhanced signaling.

## 5. Conclusions

In this study, we compared the proliferation rate, differentiation potential into three major subtypes, and mitochondrial energy metabolism between two types of porcine MSCs, namely, ADSCs and WJ-MSC, during early and late passages. The differentiation potential of ADSCs and WJ-MSCs persisted during culture. ADSCs exhibited a prominent tendency to differentiate into the adipogenic and chondrogenic lineages, whereas WJ-MSCs were more prone to osteogenic differentiation. ADSCs showed a higher proliferation rate, basal respiration, and ATP-production-coupled respiration than those of WJ-MSCs in early passages but these differences were not significant in late passages.

Therefore, the comparison between the two types of MSCs during early and late passages has broadened the scope of understanding the potential applications of these cells based on characteristics such as proliferation rate, differentiation potential, and mitochondrial metabolism. MSCs are very useful cell types that have been used for clinical purposes in animals [57], humans [58,59], and cultured meat [60]. Recently, research on fat has gained significance in the field of cultured meat production. Our study highlights that porcine adipose-derived stem cells (ADSCs) demonstrate a remarkable efficiency in adipogenic differentiation. This finding offers valuable insights for advancing the understanding of adipogenic differentiation in cultured meat production. Additionally, comparative studies involving porcine MSCs offer insights into the suitability of tissue-derived MSCs for regenerative medicine applications. Thus, this study contributes to a deeper understanding of porcine MSC characteristics, an area that has been sparsely studied in pigs.

## Figures and Tables

**Figure 1 animals-13-02947-f001:**
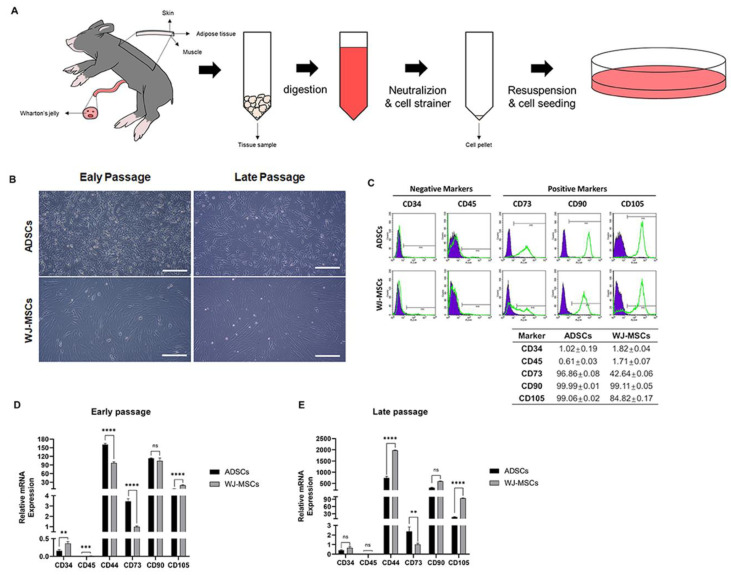
**Characterization analysis of adipose-derived stem cell (ADSC) and Wharton’s jelly MSC (WJ-MSC).** (**A**) Schematic representation depicting the process of collecting cells from both adipose tissues and Wharton’s jelly from day-1 piglets. (**B**) ADSC and WJ-MSC morphology after culturing for 72 h at passages 3 and 20. Scale bar = 200 μm. (**C**) Flow cytometry results using passages 3 MSCs for detecting negative markers CD34 and CD45 and positive markers CD73, CD90 and CD105 (n = 3). (**D**,**E**) Quantitative RT-PCR analysis was performed on ADSCs and WJ-MSCs to assess the expression of MSC-specific cell surface genes. Error bars represent the mean ± SD, n = 3; ** *p* < 0.01, *** *p* < 0.001, **** *p* < 0.0001; ns, non-significant.

**Figure 2 animals-13-02947-f002:**
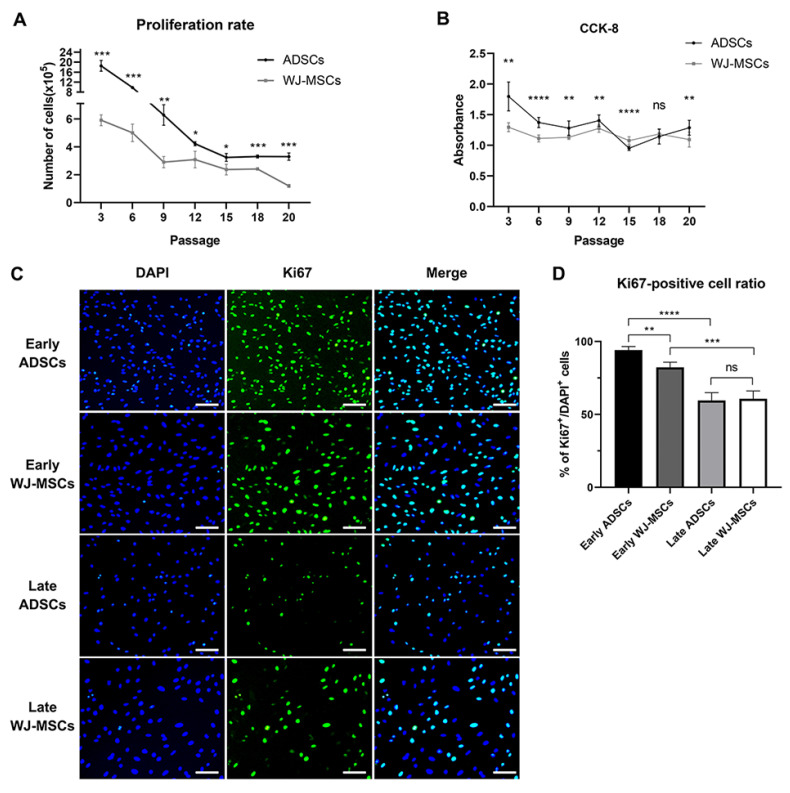
**Proliferation analysis of ADSC and WJ-MSC**. (**A**) Proliferation rate for each passage of ADSC and WJ-MSC (n = 3). (**B**) CCK-8 data of ADSCs and WJ-MSCs at different passages (n = 10). (**C**) Immunocytochemistry analysis results of ADSC and WJ-MSCs for early and late passage. GFP: Ki67; scale bar: 200 µm. (**D**) Ki67-positive cell ratio of ADSC and WJ-MSC at early and late passage (n = 4). Mean ± SD, * *p* < 0.05, ** *p* < 0.01, *** *p* < 0.001, **** *p* < 0.0001; ns, non-significant.

**Figure 3 animals-13-02947-f003:**
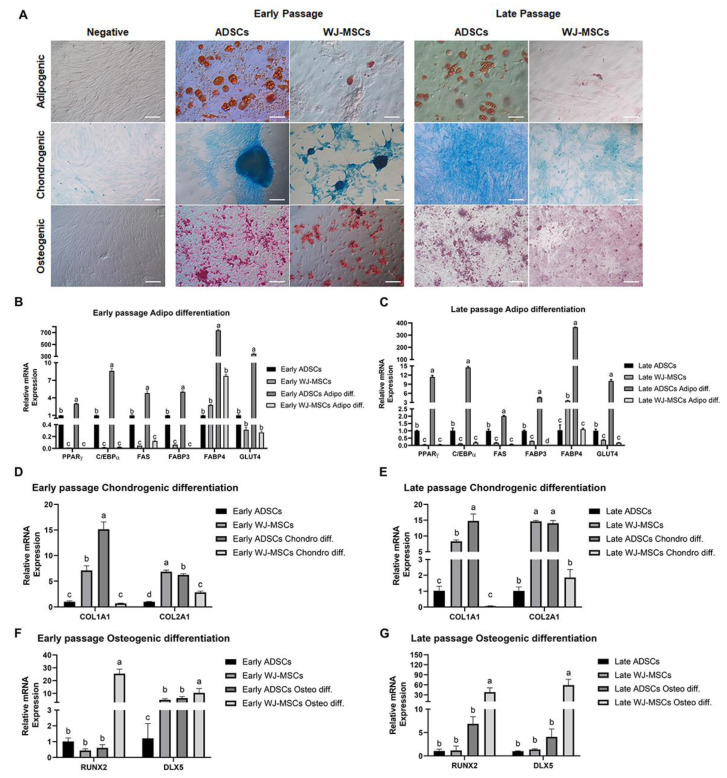
**Differentiation and staining of ADSC and WJ-MSC using, RT-qPCR**. (**A**) Verification of MSC differentiation through staining. Adipogenic, chondrogenic, and osteogenic differentiation were verified using Oil Red O, Alcian blue, and Alizarin red solution, respectively. Undifferentiated MSCs did not stain with any of the staining solutions. Scale bar: Adipogenic, 50 µm; Chondrogenic and Osteogenic, 200 µm. (**B**,**C**) Quantitative RT-PCR analyses were performed on early- and late-passage of ADSCs and WJ-MSCs to assess the expression of adipogenic differentiation markers. (**D**,**E**) Quantitative RT-PCR analyses were performed on early- and late-passage ADSCs and WJ-MSCs to assess the expression of chondrogenic differentiation markers. (**F**,**G**) Quantitative RT-PCR analyses were performed on early- and late-passage ADSCs and WJ-MSCs to assess the expression of osteogenic differentiation markers. Error bars represent mean ± SD. ^a–d^ Different letters represent significant differences (n = 3, *p* < 0.01).

**Figure 4 animals-13-02947-f004:**
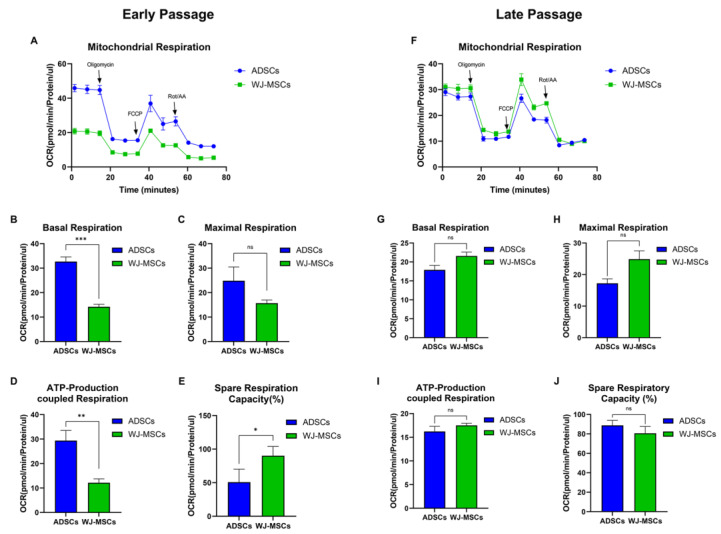
**Oxygen consumption rate (OCR) analysis of ADSCs and WJ-MSCs.** (**A**) Measurement of OCR in early ADSCs and WJ-MSCs using Seahorse XFp analyzer. (**B**–**E**) Measurement of early ADSC and WJ-MSC passage cells for (**B**) basal respiration, (**C**) maximal respiration, (**D**) ATP-production-coupled respiration, and (**E**) spare respiratory capacity (%). (**F**) Measurement of oxygen consumption rate (OCR) in late ADSC and WJ-MSC using Seahorse XFp analyzer. (**G**–**J**) Measurement of late ADSC and WJ-MSC passage cells for (**G**) basal respiration, (**H**) maximal respiration, (**I**) ATP-production-coupled respiration, and (**J**) spare respiratory capacity (%). Mean ± SD, n = 6 * *p* < 0.05, ** *p* < 0.01, *** *p* < 0.001; ns, non-significant.

**Table 1 animals-13-02947-t001:** Primers used for RT-qPCR.

Primer Name	Primer Sequence	Product Size (bp)	NCBI
GAPDH	F: ACCCAGAAGACTGTGGATGG	79	XM_021091114.1
	R: AAGCAGGGATGATGTTCTGG		
β-actin	F: GCAAGAGAGGCATCCTGACC	182	XM_021086047.1
	R: GGTCATCTTCTCACGGTTGGC		
CD34	F: GAACCGTCGCAGTTGGAGC	198	NM_214086.1
	R: GGTTGCCTCGCTGAATGGC		
CD44	F: ATGGTCGCTACAGCATCTCG	264	XM_021085286.1
	R: CTTCAGGTGGAGCTGATGCA		
CD45	F: CTGAAGACCCTCACCTGCTC	226	XM_003130596.6
	R: GCC TCC ACC TGA ACC ATC AG		
CD73	F: GAGAACCTGGCTGCTGTGT	411	XM_001927095.4
	R: CCGACCTTCAACTGCTGGAT		
CD90	F: CAG AAG GTG ACC AGC CTG AC	176	XM_013979447.2
	R: GTT CGA GAG CGG TAG GAG TG		
CD105	F: GTAGCACCAACCACAGCATCG	128	NM_214031.1
	R: CTGCTCAGTCTCTCCTGCTG		
PPARγ	F: CGACCACTCCCACTCCTTTGAC	172	XM_005669788.3
	R: CACAGGCTCCACTTTGATGGCA		
C/EBPα	F: GCAGGCAAAGCCAAGAAGTCG	143	XM_003127015.4
	R: GTCAGCTCCAGCACCTTCTGT		
FAS	F: GTCCTGCTGAAGCCTAACTC	206	NM_001099930.1
	R: TCCTTGGAACCGTCTGTG		
FABP3	F: ATGGAGGCAAACTTGTCCAC	98	NM_001099931.1
	R: ATGGGTGAGTGTCAGGATGAG		
FABP4	F: CTGGTACAGGTGCAGAAGTGG	107	NM_001002817.1
	R: CTGGTAGCCGTGACACCTT		
GLUT4	F: GCTGCCTCCTACGAGATGCT	145	NM_001128433.1
	R: TGGCCAGCTGGTTGAGTGT		
COL1A1	F: GTGTCTGCGACAACGGCAATG	240	XM_021067155.1
	R: GAAGTCCAGGTTGTCCAGGGA		
COL2A1	F: GCAACTGGGACCAAAGGGAC	113	XM_021092611.1
	R: CACCTCTGGGTCCTTGTTCAC		
RUNX2	F: CAGCCTCTTCAGCACAGTGAC	119	XM_005666074.3
	R: GGCTCACGTCGCTCATCTTG		
DLX5	F: CCGAGGTGAGAATGGTGAACGG	165	NM_001159660.1
	R: GTGTTTGCGTCAGTCCCAGC		

## Data Availability

The data supporting the findings of this study are available from the corresponding author upon reasonable request.

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
