# Peer review of "Comparative Analysis of Porcine Adipose- and Wharton’s Jelly-Derived Mesenchymal Stem Cells"

_animals, 2023, doi:10.3390/ani13182947_

Round 1

Reviewer 1 Report

Dear Authors,

I have read your manuscript and included some comments regarding minor lacks of information or additional questions (please find them in attached pdf).

The most important is the interpretation of FACS results, where the chart shows decreased expression of CD73 in WJ-MSCs while the text describes it as high and comparable expression to ADMSC. 

In general, the study is interesting for scientists working in the MSC field and shows that pigs can be also considered as raw material donors to produce novel medicines or obtain meat in vitro. Thus, it would be interesting to determine what is the difference between these populations regarding their immunomodulatory properties (medicine) and/or the myogenic differentiation potential (food) in the future studies.

In my opinion Your work is worth publishing in this journal, however minor corrections would be required.

Best regards,

I only found some minor spelling errors and single typos - marked in the pdf file.

Author Response

Response to Reviewer #1

I have read your manuscript and included some comments regarding minor lacks of information or additional questions (please find them in attached pdf).

à  I will respond to your comments in the PDF file here.

Introduction

Line 43 which->and they?

ANS : The sentence has been edited to make it clearer.

Line 57 typo?

ANS : We have corrected the typo.

Line 58 easy proliferation->what does this shortcut mean?

ANS : We have change the word to “easy expansion”

Materials and Methods

Line 72 How many piglets? How many biological repeats were done?

ANS : We have added this information to the main text. “ADSC, cells were extracted from the back fat of three 1-day-old piglets, and for Wharton's jelly, cells were extracted from more than 20 samples.”

Line 80 1100rpm-> Could Authors provide g force which is independent from centrifuge diameter?

ANS : We have provide g force values ​​for rpm values.(1100rpm= 250×g)(line81)

Line 84 We cultured cells ->Cells were cultured

ANS : We have change the sentence as per your suggestion.

Line 85 How was passaging done? What was the reduction rate of cells? 1/4? (of harvested cells went to new vessels) 1/10? Basing on this information, the cumulated population doublings could be determined, as passage number is not very specific parameter for assessing cell age, in contrast to population doublings

ANS : Based on your feedback, we have added passaging methods to M&M sections and the density of cells replated during passage.(line 85-87)

Line 85 we termed the ->were termed?

ANS : We have changed the sentence as per your suggestion.

Line 88 Could Authors provide clone numbers for monoclonal antibodies used for this measurement (if any)?

ANS : Below, we present information regarding CD clone numbers. And we have revised the clone number in the paper.(line 96, 97)

CD34, CD73, CD90, (clone number : I was unable to locate any information on the sheet, and therefore, I requested the manufacturer to provide me with the details. )

CD45(clone number : 4E9B2), CD105(MEM-229)

Line 103 analysis

ANS : We have corrected the typo.

Line 105 Were they biological or technical repeats?

ANS : It was technical replicates.

Line 114 I suppose that the reaction solution was transferred to new plate as the growth plate should be covered with cells which would influence on absorbance, am I right?

ANS : In the CCK-8 protocol provided by the manufacturer, the reagent is measured by adding the reagent to the dish in which the cells were cultured, rather than transferring the reaction solution to a new dish. In some experiments to measure survival rate, in order to reduce the influence of cells, the supernatant reacted with the reagent is removed separately and the absorbance is taken and compared (1), but there was no mention of this in the manual of CCK-8 used in this experiment. In another study utilizing the same manufacturer's CCK-8 solution, it was verified that the absorbance was assessed utilizing the plate containing seeded cells (2,3,4).

  1. Liu, H., Liu, R., Ullah, I., Zhang, S., Sun, Z., Ren, L., & Yang, K. (2020). Rough surface of copper-bearing titanium alloy with multifunctions of osteogenic ability and antibacterial activity. Journal of Materials Science & Technology, 48, 130-139.
  2. Yang, A., Peng, F., Zhu, L., Li, X., Ou, S., Huang, Z., ... & Kong, Y. (2021). Melatonin inhibits triple-negative breast cancer progression through the Lnc049808-FUNDC1 pathway. Cell Death & Disease, 12(8), 712.
  3. Yang, J., Mo, J., Dai, J., Ye, C., Cen, W., Zheng, X., ... & Ye, L. (2021). Cetuximab promotes RSL3-induced ferroptosis by suppressing the Nrf2/HO-1 signalling pathway in KRAS mutant colorectal cancer. Cell death & disease, 12(11), 1079.
  4. Bazhabayi, M., Qiu, X., Li, X., Yang, A., Wen, W., Zhang, X., ... & Liu, P. (2021). CircGFRA1 facilitates the malignant progression of HER‐2‐positive breast cancer via acting as a sponge of miR‐1228 and enhancing AIFM2 expression. Journal of Cellular and Molecular Medicine, 25(21), 10248-10256.

Line 117 How many biological and/or technical repeats were made?

ANS : We conducted a total of four biological and four technical replicates as part of the Ki67 immunocytochemistry process.

Line 128 How many biological and/or technical repeats were made?

ANS : We performed three biological replicates

Line 137 How long this process took in each case? Was it comparable between groups or repeats?

ANS : Adipogenic differentiation was evaluated over a 21-day period, Chondrogenic differentiation was carried out for 21 days, and osteogenic differentiation was performed for 14 days.

The experiment was conducted with repetition, consistently yielding identical results, thereby highlighting the discernible distinctions between the ADSCs group and the WJ-MSC group.

Following your suggestion, We have added detailed about each differentiation method.

Line 156 Why such difference?

ANS : According to the Seahorse XF Metabolism protocol, we aimed to achieve approximately 90% confluency 24 hours after seeding cells. To determine the appropriate cell concentration for each group, we seeded cells in 96 wells at various concentrations (e.g., 10,000, 15,000, 20,000, 25,000, 30,000, 35,000 cells/well) prior to conducting the metabolism experiment. Additionally, to account for potential errors related to cell numbers, we performed protein quantification using RIPA buffer for each well following the metabolism experiment.

RESULT

Line 173 Figure is of low resolution and it is hard to see descripted traits

ANS : Following your suggestion, we have adjusted the resolution to improve figure quality, and a separate figure file have provided. If you still have low resolution problem, the figure quality may be decreased after converting them to the revision version. Rest assured, the published paper will showcase the high-quality figures as originally intended.

Line 175 As above

ANS : The resolution has been adjusted to improve figure quality. If you still have low resolution problem, the figure quality may be decreased after converting them to the revision version.

Line 176 Results show that CD73 is more than 2 times lower for WJ-MSC than ADMSC...

ANS : Thanks for your query. In some studies, There are even differences in the expression of CD markers between WJ-MSCs from different individuals (1). In addition, WJ-MSCs and ADSCs, there are differences in the expression of CD markers for each passage, and CD markers sometimes increase and then decrease (3). In another case, it is observed that CD105 expression tends to decrease in MSCs extracted from individuals with obesity traits compared to those with normal traits (2). Thus, the difference in the expression of CD surface markers in our results is attributed to the different origins of MSCs.

  1. Pham, H., Tonai, R., Wu, M., Birtolo, C., Chen, M., (2018). CD73, CD90, CD105 and Cadherin-11 RT-PCR Screening for Mesenchymal Stem Cells from Cryopreserved Human Cord Tissue. Int J Stem Cells 11, 26-38. doi:10.15283/ijsc17015.

  1. Wu, C. L., Diekman, B. O., Jain, D., & Guilak, F. (2013). Diet-induced obesity alters the differentiation potential of stem cells isolated from bone marrow, adipose tissue and infrapatellar fat pad: the effects of free fatty acids. International Journal of obesity, 37(8), 1079-1087.

  1. Hendrijantini, N., & Hartono, P. (2019). Phenotype characteristics and osteogenic differentiation potential of human mesenchymal stem cells derived from amnion membrane (HAMSCs) and umbilical cord (HUC-MSCs). Acta Informatica Medica, 27(2), 72.

Line 178 Seems that CD73 is significantly lower

ANS : This problem was addressed in the line 176.

Line 181 Data do not show this - there is a clear view that expression of CD73-coding gene in WJ-MSCs is not increased at all, which corresponds with the results from FACS in this case, when compared to expression of other genes coding much more positive

ANS :

“This was corroborated by the results of the RT-qPCR analysis of MSC markers, where the negative markers (CD34 and CD45) were not expressed whereas positive markers (CD44, CD73, CD90, and CD105) were highly expressed (Figure 1C).”

In these paragraph, this means that positive markers are relatively higher than negative markers. And the difference in expression of CD73 was explained above in the line 176.

Line 182 Where the FACS data comparison between early and late passages can be seen? Which populations are shown on 1B?

ANS : We have corrected that sentence. In our study, FACS data was necessary to verify the characteristics of ADSC and WJ-MSC in early passage.

Line 188 Cannot find two groups on figure...

ANS : We have corrected the figure legend.

Line 197 Graph CCK-8 show that there is high significance in differences between groups at passage 15...

ANS : Following your suggestion, I have objectively revised the sentence.

Line 201 What was the methodology to measure and prepare the graphs with quantfied protein expression? There is no single word about it in the M&M section

ANS: In our study, protein quantification was not plotted for Ki67. The number of ki67-positive cells was counted and quantified by dividing by the number of DAPI-positive cells.

Line 220 understand?

ANS : We have corrected the sentence to ”To gain a better understanding of~ “

Discussion

Line 304 Results showed about 50% of positivity for WJ MSC...

ANS : We have corrected that sentence.

Line 315 This statement is very subjective, what does it mean considerably high?

ANS : We have revised that sentence to “Here, we observed that the proliferation rates of ADSCs and WJ-MSCs were significantly higher in early passages compared to late passages.“ (line 325-326)

Line 371 The most important property of MSCs in clinical setting is their immunomodulatory capacity - it would be very interesting to compare this parameter between these two cell populations in the future study

ANS : Thank you for this comment. This top

Reviewer 2 Report

Overall Review:

This paper presents a comparative analysis of porcine adipose-derived stem cells (ADSCs) and Wharton's jelly-derived mesenchymal stem cells (WJ-MSCs) in terms of their proliferation rate, differentiation potential, and mitochondrial metabolism. The study aims to provide insights into the unique properties of porcine MSCs, which are less studied compared to other animal species. The findings of this study can contribute to a better understanding of porcine MSCs and their potential applications in various fields. This manuscript may be accepted after minor modification.

Questions to Authors and Suggestions for Rebuttal:

1) Have you considered comparing the findings of this study with previous research on porcine MSCs or MSCs from other animal species? This would provide a better context for the significance of the findings.

2) Could you provide more insights into the molecular mechanisms underlying the observed differences in proliferation rate, differentiation potential, and mitochondrial metabolism between ADSCs and WJ-MSCs?

3) How do the findings of this study contribute to the field of regenerative medicine and cultured meat research?

no

Author Response

1) Have you considered comparing the findings of this study with previous research on porcine MSCs or MSCs from other animal species? This would provide a better context for the significance of the findings.

ANS: Thank you for your comments. To emphasize the significance of our finding, we have added previous studies on other species (line 343-346).

2) Could you provide more insights into the molecular mechanisms underlying the observed differences in proliferation rate, differentiation potential, and mitochondrial metabolism between ADSCs and WJ-MSCs?

ANS: To emphasize the importance of our research, we have added the content to the discussion section (line 358-361 , line372-378, line 381-385).

3) How do the findings of this study contribute to the field of regenerative medicine and cultured meat research?

ANS: Following your suggestion, we have added the content to the Conclusion section (Line 399-405).

Reviewer 3 Report

The author aimed to compared two types of porcine mesenchymal stem cells (MSCs) isolated from the dorsal subcuta- 24 neous adipose tissue [adipose-derived stem cells (ADSCs)] and Wharton's jelly of the umbilical cord 25 [Wharton's jelly-derived mesenchymal stem cells (WJ-MSCs)] of 1-day-old piglets.

The manuscript is well written. I have only a few suggestions for the author the improve the manuscript.

Methods

I suggest a schematic representation of the sample collection the cells of adipose tissue and jelly.

Results 

Insert the description the scale bar, figure 2 and 3 line 204 and 248.

Author Response

The author aimed to compared two types of porcine mesenchymal stem cells (MSCs) isolated from the dorsal subcutaneous adipose tissue [adipose-derived stem cells (ADSCs)] and Wharton's jelly of the umbilical cord [Wharton's jelly-derived mesenchymal stem cells (WJ-MSCs)] of 1-day-old piglets.

The manuscript is well written. I have only a few suggestions for the author the improve the manuscript.

Methods

I suggest a schematic representation of the sample collection the cells of adipose tissue and jelly.

ANS: Based on your thoughtful suggestions, we have included a visual representation demonstrating the process of extracting mesenchymal stem cells from Wharton's jelly of umbilical cord and the dorsal subcutaneous adipose tissues from a 1-day-old piglet (Figure 1A).

Results

Insert the description the scale bar, figure 2 and 3 line 204 and 248

ANS: We have included scale bars as per your suggestion.